# Identification of a Novel p53 Modulator Endowed with Antitumoural and Antibacterial Activity through a Scaffold Repurposing Approach

**DOI:** 10.3390/ph15111318

**Published:** 2022-10-26

**Authors:** Elisa Nuti, Valeria La Pietra, Simona Daniele, Doretta Cuffaro, Lidia Ciccone, Chiara Giacomelli, Carolina Cason, Alfonso Carotenuto, Vincenzo Maria D’Amore, Eleonora Da Pozzo, Barbara Costa, Riccardo Di Leo, Manola Comar, Luciana Marinelli, Claudia Martini, Armando Rossello

**Affiliations:** 1Department of Pharmacy, University of Pisa, 56126 Pisa, Italy; 2Department of Pharmacy, University of Naples Federico II, 80131 Naples, Italy; 3Unit of Advanced Microbiology Diagnosis and Translational Research, Institute for Maternal and Child Health-IRCCS “Burlo Garofolo”, 34137 Trieste, Italy; 4Department of Medical, Surgical and Health Sciences, University of Trieste, 34127 Trieste, Italy

**Keywords:** p53 modulator, antibacterial activity, glioblastoma multiforme, p53–MDM2 complex, *Chlamydia trachomatis*, virtual screening

## Abstract

Intracellular pathogens, such as *Chlamydia trachomatis*, have been recently shown to induce degradation of p53 during infection, thus impairing the protective response of the host cells. Therefore, p53 reactivation by disruption of the p53–MDM2 complex could reduce infection and restore pro-apoptotic effect of p53. Here, we report the identification of a novel MDM2 inhibitor with potential antitumoural and antibacterial activity able to reactivate p53. A virtual screening was performed on an in-house chemical library, previously synthesised for other targets, and led to the identification of a hit compound with a benzo[a]dihydrocarbazole structure, RM37. This compound induced p53 up-regulation in U343MG glioblastoma cells by blocking MDM2–p53 interaction and reduced tumour cell growth. NMR studies confirmed its ability to dissociate the MDM2–p53 complex. Notably, RM37 reduced *Chlamydia* infection in HeLa cells in a concentration-dependent manner and ameliorated the inflammatory status associated with infection.

## 1. Introduction

The tumour suppressor protein p53 has a well-known role in regulating genomic stability and response to DNA damage in normal cells. Its functions include regulation of cell cycle arrest, induction of apoptosis, and cellular senescence in order to block the proliferation of damaged cells [1,2]. p53 mainly functions as a transcription factor by binding to specific DNA sequences and by activating or repressing a large number of target genes in response to oxidative stresses, expression of oncogenes, or hypoxia conditions [3,4]. In more than half of all human cancers, the gene that encodes p53 (*TP53*) is inactivated or mutated, thus producing the loss of p53 tumour suppressor function [5,6]. Its levels are tightly controlled in normal cells by an oncoprotein called HDM2 (Human Double Minute 2, also known as MDM2), which inactivates p53 by direct binding to the *N*-terminal transactivation domain or by activating proteasomal degradation via ubiquitination [7,8]. Loss of p53 activity, by mutation or MDM2 overexpression, is the most common event in cancer development, and, in the last decade, many studies have been conducted to identify small-molecule inhibitors of MDM2−p53 interaction as a therapeutic strategy for cancer treatment [9,10,11]. In particular, Nutlin-3a [12] (Figure 1) is a 4,5-dihydroimidazoline discovered by Roche, able to reactivate p53 signalling in cancer cells acting as an MDM2 inhibitor.

Recently, several bacterial pathogens have been shown to inactivate p53 during infection [13]. In fact, DNA damage caused by many bacterial infections stimulates p53, which triggers a protective response of the host cells eventually leading to apoptosis. Intracellular pathogens, similarly to tumour cells, need to obtain energy and nutrients from their host for successful growth. Therefore, bacteria such as *Shigella flexneri* [14], *Helicobacter pylori* [15], and *Chlamydia trachomatis* [16] have been shown to induce degradation of p53 through MDM2 activation to delay host cell death, favouring their replications. In particular, the dependence of the pathogen’s growth on a functional MDM2–p53 axis highlights a potentially carcinogenic role for *C. trachomatis* [17] and *H. pylori* infections.

On the basis of these findings, in the present paper, we described our search for novel MDM2 inhibitors able to restore p53 activity with the primary goal of fighting bacterial infections. Following a scaffold repurposing approach, we performed a virtual screening (VS) campaign on a focused in-house chemical library composed of small molecules synthesised over the years for different therapeutic targets. The most promising compounds were tested in vitro to confirm whether they showed a detectable inhibition of MDM2. A hit compound with a benzo[a]dihydrocarbazole structure turned out to be the most interesting one and was assayed on three different glioblastoma multiforme (GBM) cell lines (U343MG, U87MG, and T98G) to prove its activity on p53–MDM2 complex, its ability to restore p53 function, and its capability to reduce the GBM cell growth. Finally, its inhibitory activity on chlamydial growth was evaluated in comparison with Nutlin-3a. 

## 2. Results and Discussion

### 2.1. Virtual Screening Campaign

Looking at aryl-indole-based MDM2 inhibitors that we recently discovered (**1**, Figure 1) [18,19] and comparing them with other well-known inhibitors (**2**–**5**) [12,20,21,22], we observed that they are all developed around a nitrogen-containing heterocyclic scaffold. Thus, with the aim of finding novel small molecules that can target MDM2 protein, we virtually screened our focused library composed of more than 220 molecules containing carbazoles, hexahydrocycloheptaindoles, dihydronaphthalen-hydrazine, and other similar scaffolds. In fact, these molecules share similar chemical features with the aforementioned inhibitors (Figure 1). Virtual screening calculations were carried out by means of Glide5.5 (Glide, version 5.5; Schrodinger, LLC: New York, 2009) on an MDM2 3D structure (PDB code: 3LBL; see Methods section for structure choice criteria). 

Docking results of such a library into the canonical MDM2 pocket were sorted on the basis of the docking scores, which ranged from −8.229 to −0.343. Solutions with a docking score lower than the average docking score calculated for the known active compounds (−6.00) were retained (145 molecules; see Methods for details) and visually inspected into the MDM2 binding site. Only the molecules that are able to occupy at least two of the three MDM2 subpockets (respectively interacting with the three critical residues Leu26, Trp23, and Phe19 of p53) [23] were taken into account (see Figure 2 for representative binding modes).

As a result, six compounds were selected for in vitro tests with the aim to evaluate their ability to inhibit MDM2 (expressed in % inhibition at 20 µM; Table 1). 

Interestingly, two benzo[a]dihydrocarbazole derivatives, **RM37** [24] and **RM58**, were found able to induce a satisfactory inhibition (% inhibition > 70%) against MDM2. Thus, we decided to carry on NMR studies to confirm that the most active compound, **RM37**, could disrupt the p53–MDM2 complex. 

### 2.2. NMR Studies 

Holak et al. have developed an NMR assay to determine the ability of antagonists to dissociate protein–protein complexes. The method was named AIDA (for antagonist-induced dissociation assay) [26,27]. The AIDA can work with a complex made by a large protein fragment (larger than 30 kDa) and a small reporter protein (less than 20 kDa). The AIDA makes use of two-dimensional ^15^N-HSQC spectra; however, in the presence of flexible protein residues, 1D proton NMR spectra may be sufficient for monitoring the states of the complex upon addition of ligands. Since the *N*-terminal domain of p53 is highly flexible, the MDM2–p53 complex is suitable for 1D proton NMR application.

Particularly, the signals from ^N^H^ε^ side chains of W23 and W53 are sharp in the free p53 1D proton spectrum. After the complex formation, the W23 signal disappears, since W23 comprises the primary binding site for MDM2. In fact, residues 17–26 of p53 participate in well-defined structures of large MDM2–p53 complexes. In contrast, W53 is still not structured when p53 is bound to MDM2. Due to the reduced flexibility in the complex, the observed 1/T2 transverse relaxation rate of the bound W23 significantly increases, leading to the broadening of NMR resonances and in the disappearance of this signal from the spectra. 

^1^H NMR signals of the tryptophan residues of the MDM2–p53 complex are shown in Figure 3a (0.1 mM, only the W53 ^N^H^ε^ side chains signal can be detected). 

After the addition of **RM37** (0.2 mM final concentration) to the MDM2–p53 complex, the W23 peak appeared (Figure 3b), thus proving the ability of **RM37** to dissociate the complex. Nutlin-3a was also used as a positive control (Figure 3c), causing a complete p53 release. Thus, we can claim that **RM37** dissociated the p53–MDM2 complex but was not as efficient as Nutlin-3a, causing about 60% p53 release. The full spectra of the three complexes are reported in the Appendix A.

### 2.3. p53 Stabilisation and Reactivation of p53 Pathway in GBM Cells

The cytotoxic effects of **RM58** and **RM37** were firstly analysed in U343MG cells as a representative GBM cell line that overexpresses the murine double minute 2 (MDM2) and maintains wild-type p53 [28]. Nutlin-3a was tested as a reference p53–MDM2 inhibitor effective in GBM tumour cells [29]. **RM58** did not significantly alter the number of living cells and the percentage of dead cells (Figure 4a,b).

Conversely, **RM37** was able to significantly counteract the U343MG cell proliferation reducing the number of living cells (Figure 4a). In parallel, it caused a significant increase in dead cells when tested at the highest concentration (20 µM; Figure 4b). Interestingly, **RM37** affected the GBM cell growth similarly to Nutlin-3a tested at 10 µM (used as a positive control), corresponding to about 100-fold of its IC_50_ (108.0 ± 4.5 nM) calculated in similar dissociation experiments [19]. Based on these results, the activity of **RM37** on p53 stabilisation was deeply investigated. First, the ability of **RM37** to modify the dissociation of the pre-formed MDM2–p53 complex was evaluated (Figure 4c) using a direct quantitative sandwich immuno-enzymatic ELISA assay [19]. **RM37** effectively dissociated the MDM2–p53 complex in the crude lysates of GBM cells with an IC_50_ value of 222 ± 40 nM. 

Then, additional experiments were performed in U343MG cells using a concentration of 100-fold IC_50_. The effect of **RM37** on p53 accumulation in U343MG cells was examined. Challenging cells with the compound for 4, 8, and 10 h led to a time-dependent increase in p53 protein levels (Figure 5). 

The accumulation of p53 became significant after 8 h of treatment (Figure 5). As a comparison, Nutlin-3a induced a significant accumulation of p53 protein after 8 h and 12 h of treatment [30]. These data suggest that **RM37** stabilises p53 protein levels, possibly by reducing MDM2-mediated p53 degradation [30].

The stabilisation of p53 can be ascribed also to an enhancement of p53 gene transcription and subsequent protein translation [30]; thus, additional experiments were performed (Figure 6). 

Until 16 h of cell treatment, **RM37** (20 µM) did not significantly modify the p53 mRNA level (Figure 6a). To confirm that the p53 accumulation was due to its decreased interaction with MDM2, the ability of **RM37** (20 µM) to dissociate MDM2 from p53 was verified in U343MG cells, which were treated for 8 h, a time of treatment in which **RM37** did not promote p53 transcriptional activity (Figure 6a). In co-immunoprecipitation experiments, only minimal amounts of p53 could be detected in MDM2 immunoprecipitates following 8 h of treatment (Figure 6b,c). Altogether, these results indicate that **RM37** is able to induce p53 up-regulation by blocking the MDM2–p53 interaction.

### 2.4. p53 Functional Reactivation Induced by RM37

The increase in p53 intracellular levels is linked to different effects on cell growth. In particular, the p53 functional reactivation has been widely related to the modulation of specific gene transcription [31]. To verify whether the p53 functionality was effectively reactivated after a prolonged exposure time, U343MG cells were treated with **RM37** for 24 h, and the mRNA levels of MDM2, p21, and the apoptosis regulator Bax were quantified. The compound induced a significant induction of MDM2 and p21 mRNA levels (Figure 7a).

No significant effect was noticed on Bax mRNA levels. Similar results were shown by Nutlin-3a, even with a smaller extent of gene induction (Figure 7b). 

Then, the effects of p53 release from the complex with MDM2 on apoptosis induction (Figure 7c) and cell cycle blockade (Figure 7d) were evaluated. Challenging U343MG cells with RM37 (1–20 µM) for 48 h caused a significant increase in the percentage of early apoptotic cells (*p* < 0.001, versus the CTRL). This result is in accordance with the increase in dead cells after the RM37 treatment (Figure 4b) and the slight increase in Bax transcription (Figure 7a). 

p53 tightly regulates the cell cycle progression modifying the activity of different cyclin proteins. In particular, the activation of the cyclin-dependent kinase (CDK) inhibitor p21 (also known as p21^WAF1/Cip1^) caused the cell cycle arrest at both G1 and G2 phases [30,32]. Therefore, the modification of the cell cycle was evaluated using a cytofluorimetric assay (Figure 7d). Challenging U343MG cells with a high concentration of **RM37** (20 µM) caused a significant increase in the number of cells in the G1/G0 phase (62.0 ± 0.3 and 76.7 ± 2.1, CTRL and **RM37** (20 µM) respectively, ** *p* < 0.01) and a concomitant decrease in the DNA content of cells in the G2/M phases (22.6 ± 0.6 and 14.0 ± 1.1, CTRL and **RM37** (20 µM) respectively, ** *p* < 0.01). These data suggest that **RM37** caused a cell cycle block in the G1/G0 phase in accordance with the ability of Nutlin-3a and other MDM2/p53 inhibitors to produce a similar effect in GBM cells [30].

Overall, these results demonstrated the ability of **RM37** to restore the functionality of p53 through the transcriptional increase in specific p53-related genes and the induction of apoptosis and cell cycle blockade. 

### 2.5. Effects of RM37 on GBM Cell Growth

One of the main hallmarks of GBM is its high heterogeneity [33]. Thus, to deeply investigate the role of p53 reactivation mediated by **RM37** on GBM cell growth control, its ability to modify the cell growth in other GBM cell lines was evaluated (Figure 8).

In particular, U87MG expressing wild-type p53 and overexpressing the MDM2 and T98G cell expressing mutated-p53 were used [28]. **RM37** caused a significant decrease in living cells (Figure 8a) with a concomitant increase in dead cells (Figure 8b). These effects were comparable with those produced by Nutlin-3a treatment. Consistently, the decrease in U87MG growth was similar to that obtained in U343MG cells presenting similar p53 status (Figure 4a,b). Conversely, **RM37** did not significantly affect the number of T98G live cells (Figure 8c). It only produced a slight effect on the amount of T98G dead cells (Figure 8d). These data support the hypothesis that the p53 reactivation may be involved in the **RM37**-mediated antiproliferative activity in GBM cells.

### 2.6. RM37 in Chlamydia Infection

#### 2.6.1. Effects of RM37 on Chlamydia Infection

Based on previous reports demonstrating that *Chlamydia* infection induces the degradation of p53 and that compounds dissociating p53 from MDM2 can reduce the bacterial pathogen propagation [17], the effect of **RM37** on the infection was tested in comparison with Nutlin-3a, used as a positive control. Since **RM37** has a 7β-(benzo[a]dihydrocarbazolyloxyacetyl)-substituted cephalosporin structure, previously identified to have antibacterial activity in similar derivatives deprotected on carboxylate function in C4 [34], at this point of our study, we decided to synthesise its analogue deprotected on the carboxylate, **RM53** (Figure 9e and Appendix A), to exclude that the antibacterial activity of the new scaffold could be due to the typical mechanism of action of the cephalosporin nucleus instead of MDM2 inhibition. We hypothesised that the presence of a *tert*-butyl ester in the C4 position of the penem scaffold should make **RM37** unable to inhibit the penicillin-binding proteins (PBPs) that are the therapeutic target of β-lactam antibiotics. In particular, HeLa cells were infected with *Chlamydia trachomatis*, in the absence or in the presence of **RM37** and **RM53**, and the copies/µL of the bacterium were detected by quantitative PCR. As depicted in Figure 9a, **RM37**, **RM53**, and Nutlin-3a induced a significant reduction of *C. trachomatis* infection (i.e., copies/µL) with respect to untreated cells (indicated as Reinfection).

These effects occurred in a concentration-dependent manner and were particularly evident with **RM37**. These data evidenced that the inhibitors of the p53–MDM2 interaction are able to reduce *Chlamydia* infection, and the major activity of **RM37** with respect to **RM53** indicated that the principal mechanism of action of these compounds is not represented by PBP inhibition.

#### 2.6.2. Cytokines and Chemokines Analysis following Chlamydia Infection

The quantification of chemokines, cytokines, and growth factors concentrations was performed on supernatants of cells 24 h post reinfection at the indicated conditions.

As depicted in Figure 9 b–d, the reinfection with *Chlamydia trachomatis* showed a significant enhancement in the concentration of interleukin IL-8 [35], RANTES [36], and IFN-γ-inducible protein 10 (IP-10) [37], thus confirming that the bacteria pathogen induced the release of inflammatory molecules. 

**RM37**, **RM53**, and Nutlin-3a significantly reduced the accumulation of RANTES and IP-10 (Figure 9c,d). Of note, **RM37** was able also to counteract the infection-induced IL-8 concentration (Figure 9b). Overall, these data demonstrate the ability of MDM2–p53 ligands, and in particular of **RM37**, to ameliorate the inflammatory status associated with *Chlamydia* infection. 

#### 2.6.3. Western Blotting Analysis of p53 following Chlamydia Infection

To verify whether **RM37**-mediated disruption of the MDM2–p53 complex could affect *Chlamydia* infection, HeLa cells were infected with *Chlamydia*, and treated with **RM37** 24 h post-infection. Following treatment, p53 levels were evaluated after the infection with C. trachomatis serovar D (ATCC VR-885). Prior to infection, **RM37** was confirmed to significantly enhance p53 protein levels in HeLa cells, too (Figure 10a,b).

Moreover, p53 levels significantly increased in the case of re-infection with the lysate derived from the treatment with **RM37**. These data suggest that the compound acts through the MDM2–p53 axis in *Chlamydia* infection. 

## 3. Materials and Methods

### 3.1. Virtual Screening

For our study, our in-house database of ~220 compounds was used. Such database was prepared using LigPrep (LigPrep, version 2.5, Schrödinger, LLC, New York, NY, 2011) generating all possible tautomeric, enantiomeric, and protonation states and keeping only those possessing good ADME properties (calculated by means of QikProp). The final database was composed of 633 molecules. 

As per the MDM2 X-ray selection, multiple 3D structures of MDM2 can be found in the Protein Data Bank (PDB). Among them, we took into consideration just the ones containing the N-terminus residues 16–24 (e.g., 3LBL, 4HBM, 4DIJ, 4JVR, 4JVE, 1T4E, 4ERF, etc.) folded into an ordered helix. Following the latter and other previously described criteria [19], the structure with PDB code 3LBL [38] (resolution: 1.60 Å) containing a spirooxindole derivative was chosen for our screening campaign. 

The protein was prepared using the Protein Preparation Wizard implemented in Maestro Suite 2011 (Maestro, version 9.0.211; Schrodinger, LLC: New York, NY, USA, 2009). During the preparation, all water molecules were deleted, hydrogen atoms added, and the complex minimised. The receptor grids were generated using the grid generation in Glide 5.5 (Glide, version 5.5; Schrodinger, LLC: New York, NY, USA, 2009) centred around the crystallised ligand using default settings. 

For the VS on the MDM2 structure, the SP mode in Glide was used first, retaining only the best 50% hits that were redocked in XP mode, letting all other settings at the default. Solutions with a docking score higher than the average docking score of the known active compounds (−6.00) were discarded. Structures and inhibition data for known inhibitors (~30 hits including compound **1**) were downloaded from the BindingDB database [39]. The molecules were prepared using LigPrep, considering all the protonation states, and were docked with Glide5.5 in XP mode into the MDM2 structure retaining all the good states. Results ranged from −8.149 to −4.115 with an average docking score of −6.132. Thus, the cut-off for the screening was set to −6.00. On the basis of this criterion, about 25% of the entries were retained (145 molecules) and visually inspected into the MDM2 binding site.

Figure 1 was rendered using PyMOL [40].

### 3.2. Protein Expression and Purification

The recombinant human His-tagged *N*-terminal region of MDM2 (residues 1–118) was obtained using an *Escherichia coli* BL21(DE3) RIL expression system based on a pET-46Ek/LIC (Novagen) derivative vector in which the coding region of the MDM2 sequence was cloned. In particular, transformed cells were grown at 37 °C in 1 L of an LB medium up to 0.6 A_600_, induced for 3 h with 0.4 mM IPTG and collected by centrifugation. Cells were suspended with 5 mL of 20 mM Tris-HCl pH 7.2, 20 mM β-mercaptoethanol, 1 mM PMSF, and 1 mM EDTA (buffer A) and disrupted by a French press. The cell extract obtained was then centrifuged at 13,000 rpm for 1.5 h to obtain the inclusion bodies as a pellet. This fraction was solubilised with 5 mL of 8 M urea in buffer A and centrifuged at 13,000 rpm for 1.5 h to remove the insoluble material. MDM2 was purified under denaturing conditions by affinity chromatography, and to this aim, the supernatant was incubated overnight at 4 °C with 1 mL of NiNTA Agarose (Qiagen, Milan, Italy). After washing of the packed resin with 8 M urea in buffer A, pure MDM2 was eluted with 250 mM imidazole in the washing buffer. Fractions containing MDM2 were pooled together (4 mL) and refolded through three dialysis steps. The protein sample was firstly dialysed against 1 L of 2 M urea in 20 mM Tris-HCl pH 7.2, 20 mM β-mercaptoethanol, and 1 mM EDTA for 2 h at room temperature. A second dialysis step was carried out against 1 M urea in 20 mM Tris-HCl pH 7.2 and 20 mM β-mercaptoethanol for 2 h at 4 °C, and finally, the protein sample was dialysed overnight at 4 °C against 20 mM Tris-HCl pH 7.2 containing 20 mM β-mercaptoethanol.

The recombinant human His-tagged p53 protein (residues 1–312) was expressed and purified using the same protocol above-mentioned for MDM2 except that the buffer used in the final dialysis contained also 0.2 mM ZnCl_2_.

Purified proteins appeared homogeneous on 12% SDS-polyacrylamide gel electrophoresis, and the protein concentration was derived from absorbance readings at 280 nm using a molar absorption coefficient (1 cm) of 0.54 and 0.79 calculated for MDM2 and p53, respectively, on the basis of their amino acid sequence.

Commercially available chemicals were purchased from Sigma-Aldrich (Milan, Italy).

### 3.3. NMR Study of p53-MDM2 Interaction

NMR spectra were acquired at 25 °C on a Varian Unity INOVA (Palo Alto, CA) 700 MHz spectrometer equipped with a cryoprobe. NMR samples contained 0.1 mM of proteins in 20 mM Tris-HCl, 150 mM KCl, pH 7.4, 5 mM β-mercaptoethanol, and 0.02% NaN_3_. Water suppression was carried out by gradient echo [41]. NMR data were processed using the Bruker program BioSpin 3.0. For NMR ligand binding experiments, 300 µL of the protein sample containing 10% D_2_O, at a concentration of about 0.1 mM, and a 10 mM stock solution of compound **RM37** in DMSO-d_6_ were used in the experiments. The final molar ratio protein/inhibitor was 1:2. Commercially available chemicals were purchased from Sigma-Aldrich (Milan, Italy).

### 3.4. Cell Culture

Human glioblastoma cell line U87MG (WHO grade IV) was purchased from the National Institute for Cancer Research of Genoa (Italy). The human T98G and U343MG cell lines (WHO grade IV) were obtained from the American Type Culture Collection (USA) and CellLines Service GmbH (Germany), respectively. Cell lines were controlled for DNA profiling. U343MG cells were maintained in Minimum Essential Medium Eagle with 2 mM l-glutamine (Sigma-Aldrich, Milan, Italy), 1.5 g/L sodium bicarbonate (Sigma-Aldrich, Milan, Italy), 10% FBS (Sigma-Aldrich, Milan, Italy), 100 U/mL penicillin, 100 mg/mL streptomycin (Sigma-Aldrich, Milan, Italy), 1% non-essential amino acids, and 1.0 mM sodium pyruvate (Sigma-Aldrich, Milan, Italy). U87MG and T98G cells were maintained in an RPMI medium supplemented with 10% FBS, 2 mM L-glutamine, 100 U/mL penicillin, 100 mg/mL streptomycin, and 1% non-essential amino acids (NEAA) (Sigma-Aldrich, Milan, Italy). HeLa cells were propagated in high glucose Dulbecco’s modified Eagle’s medium (DMEM) (Euroclone) with 10% foetal bovine serum (FBS) (Euroclone). Cell cultures were maintained on a 6-well cell culture cluster (Corning) at 35 °C with 5% CO_2_. Mycoplasma contaminations were excluded by PCR using primers GPO-3 and MGSO.

### 3.5. Cell Proliferation and Viability Assay

U343MG, U87MG, and T98G cells were seeded at a density of 1 × 10^4^ cells/cm^2^. After 24 h, the cells were treated with different concentrations of the compounds as indicated in the figure legends, and the effects on cell viability were evaluated using the trypan blue (Sigma-Aldrich, Milan, Italy) exclusion assay as previously reported [19]. Briefly, cells were incubated with an equal volume of 0.4% trypan blue dye, and the blue (dead) and white (living) cells in each well were manually counted. The number of dead cells for each condition was reported as the percentage of cells relative to the total cells in the same treatment.

### 3.6. Analysis of Native Human p53–MDM2 Complex Dissociation by In Vitro ELISA-Based Assay

The ability of the newly synthesised compound **RM37** to dissociate the native MDM2–p53 complex was assessed by a quantitative immuno-enzymatic assay, as previously reported [19]. Briefly, lysate samples of GBM cells were incubated for 10 min at room temperature with the solvent DMSO (control) (Sigma-Aldrich, Milan, Italy) or with increasing concentrations of the compound **RM37** (from 1 nM to 50 µM) before being transferred to the wells coated with the antibody anti-MDM2 (sc-965, Santa Cruz Biotechnology, 1550 in 0.05% poly-L-ornithine) (Sigma-Aldrich, Milan, Italy). Following washings to remove unbound MDM2 and BSA treatment to block nonspecific sites, each sample was incubated for 90 min with the anti-p53 antibody (sc-6243, Santa Cruz Biotechnology, 15250 in 5% milk). The levels of the MDM2–p53 complex were quantified using an HRP-conjugated antibody and the TMB substrate.

### 3.7. Determination of p53–MDM2 Complex in U343MG Cells

U343MG cells were treated with DMSO (control) with 20 µM **RM37** for 8 h. Following the treatments, the amount of the p53–MDM2 complex was determined using co-immunoprecipitation experiments [30]. 

One milligram of cell lysates was precleared with protein A-Sepharose (1 h at 4 °C) and then centrifuged for 10 min at 14,000× *g* [30]. The supernatants were incubated with an anti-MDM2 antibody (5 µg/sample) overnight at 4 °C and then immunoprecipitated with protein A-Sepharose (2 h at 4 °C). After washing, the immunocomplexes were resolved by SDS-PAGE (8.5%), transferred to PVDF membranes, and probed overnight at 4 °C with primary antibodies to p53 (FL-393, 1:500, Santa Cruz Biotechnology, Santa Cruz, CA, USA) or MDM2 (C-18, 1:500, Santa Cruz Biotechnology, Santa Cruz, CA, USA) [30]. Densitometric analysis of immunoreactive bands was performed using Image J software v 1.52 (NIH, Bethesda, MD, USA).

### 3.8. Analysis of p53 Stabilisation in U343MG Cells

U343MG cells were treated with DMSO (control) or with 20 µM **RM37** or 10 µM Nutlin-3 for 4, 8, or 10 h. Following the treatments, the cells were then lysed for 60 min at 4°C, and cell extracts (40 µg of proteins) were diluted in Laemmli solution and resolved by SDS-PAGE [19] using a specific antibody against p53 (1:300, FL-393, Santa Cruz Biotechnology, Santa Cruz, USA). β-Actin (diluted 1:5000, MAB1501, Millipore) was employed as the loading control. The primary antibody was detected using a secondary antibody conjugated to peroxidase (1:10.000, Sigma-Aldrich, Milan, Italy). The peroxidase was detected using a chemiluminescent substrate (ECL, PerkinElmer, Milan, Italy). Densitometric analysis of immunoreactive bands was performed using ChemiDoc™ XRS+ System (BioRad, Hercules, CA, USA) and ImageJ software (version 8, ImageJ.nih.gov).

### 3.9. RNA Extraction and Real-Time PCR Analysis in U343MG Cells 

U343MG cells were treated with DMSO (control) or **RM37** (10 µM or 20 µM) or Nutlin-3a (10 µM or 20 µM) for 24 h. At the end of the treatment, total RNA was extracted using an RNeasyHMini Kit (Qiagen, Hilden, Germany) and then transformed in cDNA using an i-Script cDNA synthesis kit (BioRad, Hercules, CA., USA). RT-PCRs were performed for 40 cycles using the following temperature profiles: 98 °C for 30 min, T °C annealing for 30 min and 72 °C for 3 min. Primer sequences and annealing temperatures were previously reported [30].

### 3.10. Apoptosis and Cell Cycle Analyses

For apoptosis measurement, U343MG cells (1 × 10^4^ cells/cm^2^) were treated with DMSO (CTRL) or **RM37** at different concentrations (1–20 µM) for 48 h. Then, the percentages of living, apoptotic, and dead cells were stained with Annexin V and 7-aminoactinomycin D (7-AAD) and analysed by a Muse™ Cell Analyser (Merck KGaA, Darmstadt, Germany). Similarly, the cell cycle analysis was performed after the treatment of U343MG cells (1 × 10^4^ cells/cm^2^) with DMSO (CTRL) or **RM37** at different concentrations (1–20 µM) for 48 h. The quantification of the percentage of cells in different cell phases was performed using the Muse™ Cell Analyser.

### 3.11. Chlamydia Trachomatis Infection

HeLa cells were grown to 80% confluence on a 6-well cell culture cluster and inoculated with C. trachomatis serovar D (ATCC VR-885, ATCC, Milan, Italy) at a multiplicity of infection (MOI) 0.5. Briefly, the inoculum was centrifuged with glass beads and diluted in DMEM to reach the indicated MOI at a final volume of 500 µL per well. After centrifugation at 1000 *g* for 45 min, cells were further incubated with DMEM containing 10% FBS for 24 h at 35 °C with 5% CO_2_. At the same time, uninfected cells, seeded at the same conditions, were carried on.

Twenty-four hours after infection, HeLa cells were treated with Nutlin 3-a, **RM37**, and **RM53** dissolved in DMSO and resuspended in a complete growth medium at final concentrations of 5 µM and 20 µM for 48 h. Untreated cells were carried on under the same conditions.

### 3.12. Infectious Progeny Assays

Pellets of HeLa-infected cells were collected and lysed by vortexing with glass beads. Lysates were diluted in DMEM with 10% FBS and transferred to fresh HeLa cells at a final dilution of 1:40. After centrifugation at 3000 RPM for 45 min, the cells were further incubated with a complete medium for 24 h at 35 °C and 5% CO_2_. Twenty-four hours after reinfection with lysates, supernatants and pellets were collected and stored at −80°C for analyses. Untreated cells were carried on at the same conditions.

### 3.13. DNA Isolation and Real-Time PCR Analyses for Chlamydia Trachomatis Quantification

DNA was isolated from pellets (1 × 10^5^ cells) and eluted in a final volume of 50 μL using a commercial kit (Maxwell DNA kit, Promega). DNA was stored until the time of analysis at −20 °C. The detection of *Chlamydia trachomatis* was performed in duplicate on DNA from pellets 24 h after the reinfection using the commercial kit RealLine *Chlamydia trachomatis* Fla-Format (Bioron) following the manufacturer’s instructions. PCR assays were run using an ABI PRISM 7900 Sequence Detection System real-time cycler (Applied Biosystem). A standard curve for the absolute quantification was generated using serial dilutions of the *C. trachomatis* strain (ATCC VR-885) at a known concentration. 

### 3.14. Cytokines and Chemokines Analysis

The quantification of chemokines, cytokines, and growth factor concentrations was performed on supernatants of cells 24 h post reinfection at the indicated conditions, and on uninfected cells, using quantitative Luminex Multiplex assays (Bio-Plex, BIO-RAD), according to the manufacturer’s instructions, for measurement of 48 analytes. Briefly, 50 μL of each supernatant and reaction standards was added to a 96-multiwell plate containing analyte beads followed by incubation for 30 min at room temperature. The antibody-biotin reporter was added after washing and incubated for 10 min with streptavidin–phycoerythrin. Determination of levels of the cytokines was performed by a Bio-Plex array reader (Luminex). Bio-Plex Manager software optimised the standard curves and reported the data as median fluorescence intensity (MFI) and concentration (pg/mL). The mediators, which showed variation of expression, are reported.

### 3.15. Western Blotting Analysis of p53 in HeLa Cells

Western blot analysis was performed for the evaluation of p53 protein levels. HeLa cells were lysed by the addition of 250 μL of RIPA buffer containing proteases’ inhibitors. Equal amounts of the cell extracts (20 μg of proteins) were diluted in Laemmli solution, resolved by SDS-PAGE (7.5%), transferred to PVDF membranes, and probed overnight at 4 °C with primary antibody anti-p53 (diluted 1:1000; 70R-31561, Filtzgerald), as described above.

### 3.16. Statistical Analyses

Graph-Pad Prism 6.0 (GraphPad Software Inc., San Diego, CA, USA) was used to analyse all the data. Data are presented as the means ± SEM or SD as reported in the respective figure legends. Based on the performed experiments, the appropriate statistical analysis was performed (e.g., one-way analysis of variance (ANOVA), followed by S Bonferroni’s corrected *t*-test for post hoc pairwise comparisons, and Student’s *t*-test).

### 3.17. Chemical Synthesis

Compounds **RM37** and **RM53** were synthesised as described in Appendix A.

## 4. Conclusions

The strategy of in-house chemical library repurposing has been successfully applied to our search for new molecules that are able to inhibit MDM2–p53 protein–protein interaction. 

A VS campaign led to identify the benzo[a]dihydrocarbazole derivative RM37 as a suitable candidate. Its ability to dissociate the MDM2–p53 complex was confirmed by NMR in vitro assays on the recombinant human MDM2–p53 complex. RM37’s ability to reactivate the p53 pathway was first tested in U343MG tumour cells, and the compound showed to effectively dissociate the MDM2–p53 complex in the crude lysates of GBM cells with an IC_50_ value of 222 nM. In particular, RM37 proved to restore the functionality of p53 through the transcriptional increase of specific p53-related genes and the induction of apoptosis and cell cycle arrest. After proving the efficacy of RM37 on the MDM2–p53 complex in tumour cells, this hit compound was tested on HeLa cells infected with *Chlamydia trachomatis* in comparison with Nutlin-3a.

Previous studies have reported that *Chlamydia* degraded p53 and that the pharmacological inhibition of the p53–MDM2 interaction was sufficient to disrupt the intracellular development of *Chlamydia*, interfering with the pathogen’s anti-apoptotic effect on host cells [17]. RM37 reduced *Chlamydia* infection in a concentration-dependent manner and ameliorated the inflammatory status associated with the infection. Overall, these results confirmed that restoring p53 function can be considered a new approach to treat bacterial infections in addition to traditional therapies, which, in some cases, are facing a rapid spread of antibiotic resistance. This repurposed scaffold will be subjected to further optimisation to improve its activity and pharmacokinetic properties.

## 5. Patents

Rossello, A.; Nuti, E.; Orlandini, E.; Nencetti, S.; Martini, C.; Costa, B.; Giacomelli, C.; Daniele, S. Compounds with a Benzo[a]Carbazole Structure and Use Thereof. 2019, WO2019049024A1 [42].

## Figures and Tables

**Figure 1 pharmaceuticals-15-01318-f001:**
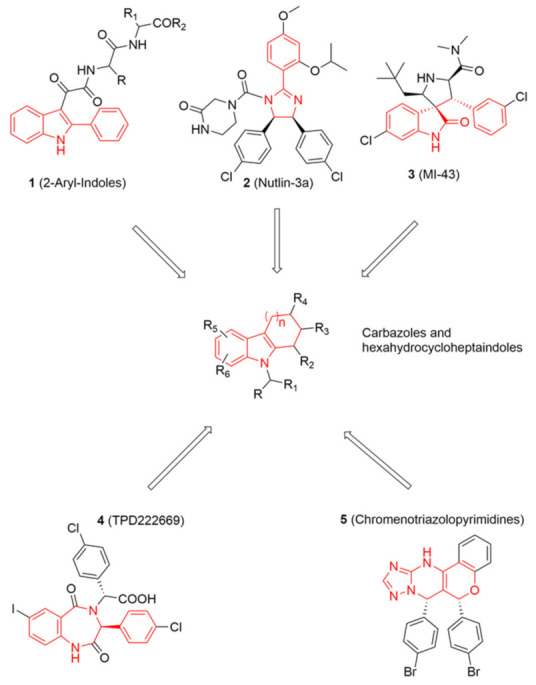
On the top and bottom of the chart, chemical structures of five known potent MDM2 inhibitors are shown. Chemical features shared among these inhibitors and some of database molecules, e.g., carbazoles (in the centre) are highlighted in red lines.

**Figure 2 pharmaceuticals-15-01318-f002:**
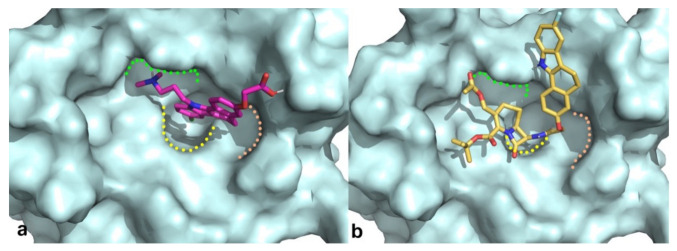
Binding modes obtained by docking results of compounds **RM58** (**a**) and **RM37** (**b**) into MDM2 binding site. Protein is shown as cyano surface, whereas **RM58** and **RM37** are shown as magenta and golden sticks, respectively. Leu26, Trp23, and Phe19 subpockets are shown with green, yellow, and pink dashed lines, respectively.

**Figure 3 pharmaceuticals-15-01318-f003:**
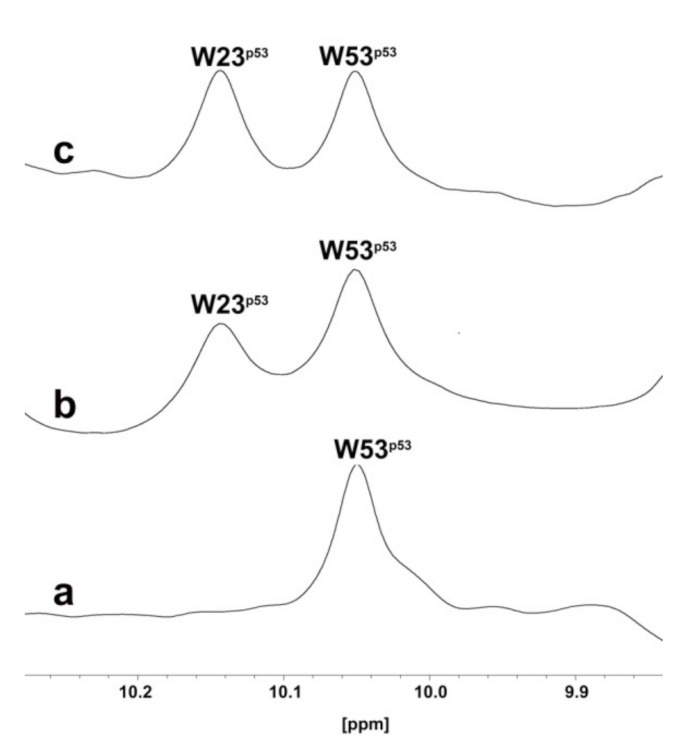
One-dimensional proton spectrum of side chains of tryptophans (W) of p53–MDM2 complex (**a**), p53–MDM2 complex after addition of **RM37** (**b**), or Nutlin-3a (**c**).

**Figure 4 pharmaceuticals-15-01318-f004:**
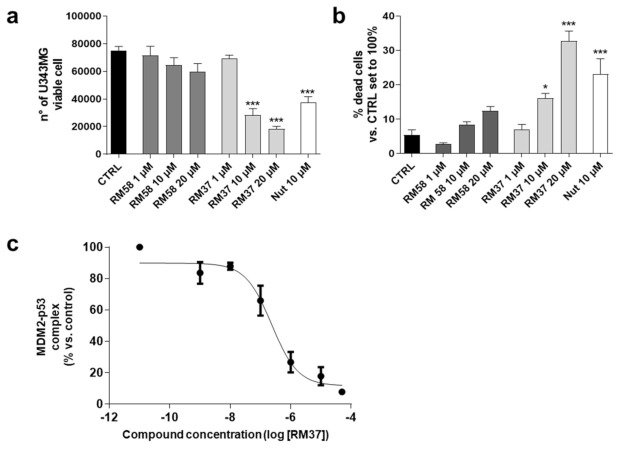
Effects of **RM37** and **RM58** on U343MG glioblastoma cells. U343MG cells were treated with different concentrations of **RM37** and **RM58** for 48h. The number of viable cells (**a**) and percentage of dead cells (**b**) were reported. Significance of the differences was determined with a one-way ANOVA with Bonferroni post-test * *p* < 0.05; *** *p* < 0.001 versus CTRL. (**c**) **RM37** was incubated with GBM cell lysates containing the native MDM2–p53 complex, and levels of MDM2–p53 complex were quantified using HRP-conjugated antibody and TMB substrate. Data represent mean ± SEM of three independent experiments.

**Figure 5 pharmaceuticals-15-01318-f005:**
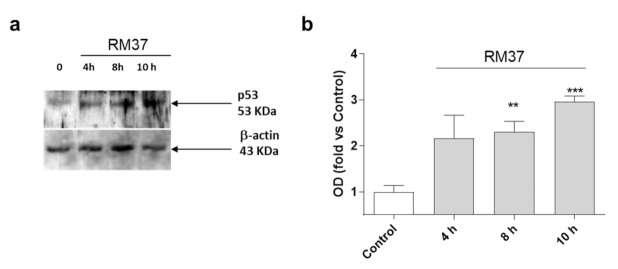
p53 protein accumulation in U343MG cells. U343MG cells were challenged with DMSO (control) or 20 µM **RM37** for 4h, 8h, or 10h. Lysates were subjected to Western blot analysis using an antibody specific for p53. One representative Western blot is shown (**a**). Full screen of Western blot assay was provided in a separate file. Quantitative analysis was performed using ImageJ and shown in panel (**b**). Data are expressed as fold of optical density (OD) of the immunoreactive band relative to that of control, set to 1, and are mean values ± SEM of three different experiments. ** *p* < 0.01, *** *p* < 0.001 versus control.

**Figure 6 pharmaceuticals-15-01318-f006:**
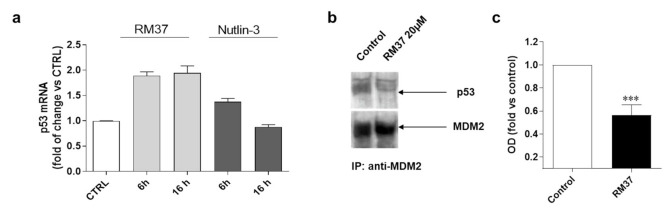
Degree of interaction of p53 with MDM2 in GBM cells. (**a**) U343MG cells were treated with 20 µM **RM37** for 6 or 16h. Relative quantification of p53 mRNA was performed by real-time RT-PCR. Data are expressed as fold of change versus control cells, set to 1. (**b**,**c**) U343MG cells were treated with 20 µM **RM37** for 8h. After incubation, immunoprecipitation assay was performed. Representative immunoblot was shown. (**b**) Full screen of Western blot assay was provided in separate file. Data are expressed as fold of optical density (OD) of immunoreactive band relative to that of control, set to 1, and are mean values ± SEM of three different experiments. *** *p* < 0.01 versus control.

**Figure 7 pharmaceuticals-15-01318-f007:**
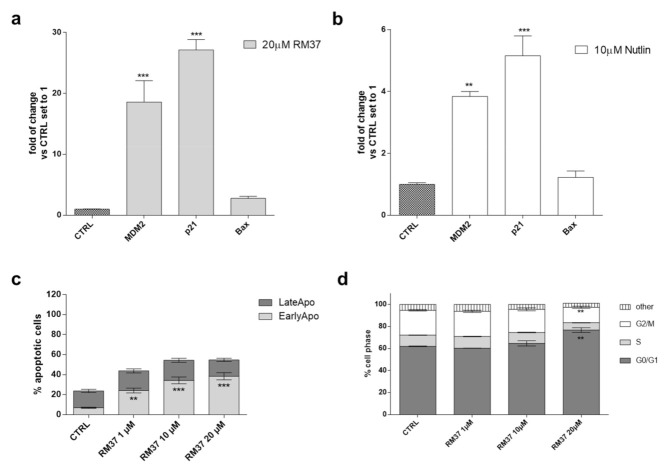
p53 functional reactivation in U343MG cells. (**a**,**b**) U343MG cells were challenged with 20 µM **RM37** or 10 µM Nutlin-3a for 24 h. Relative mRNA quantification of p53 target genes (Bax, p21, and MDM2) was performed by real-time RT-PCR, as described in the Methods section. Data are expressed as fold of change versus control cells, set to 1. (**b,c**) U343MG cells were treated with DMSO (CTRL) or different concentrations of **RM37** (1–20 µM) for 48 h. In the end, the number of apoptotic cells (**c**) and cell cycle progression (**d**) were evaluated. Data are mean values ± SEM of three different experiments. ** *p* < 0.01; *** *p* < 0.001 versus respective CTRL.

**Figure 8 pharmaceuticals-15-01318-f008:**
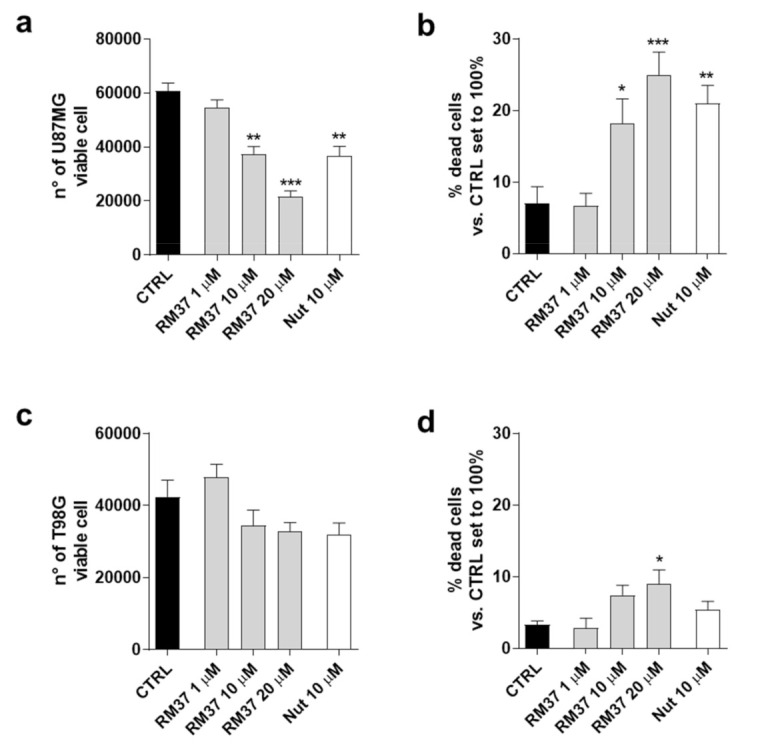
Effects of **RM37** on different glioblastoma cells. (**a**) U87MG live cells; (**b**) U87MG dead cells; (**c**) T98G live cells; (**d**) T98G dead cells following treatment with different concentrations of **RM37** (1–20 µM) for 48h. Data are mean values ± SEM of three different experiments. * *p* < 0.05; ** *p* < 0.01; *** *p* < 0.001 versus CTRL.

**Figure 9 pharmaceuticals-15-01318-f009:**
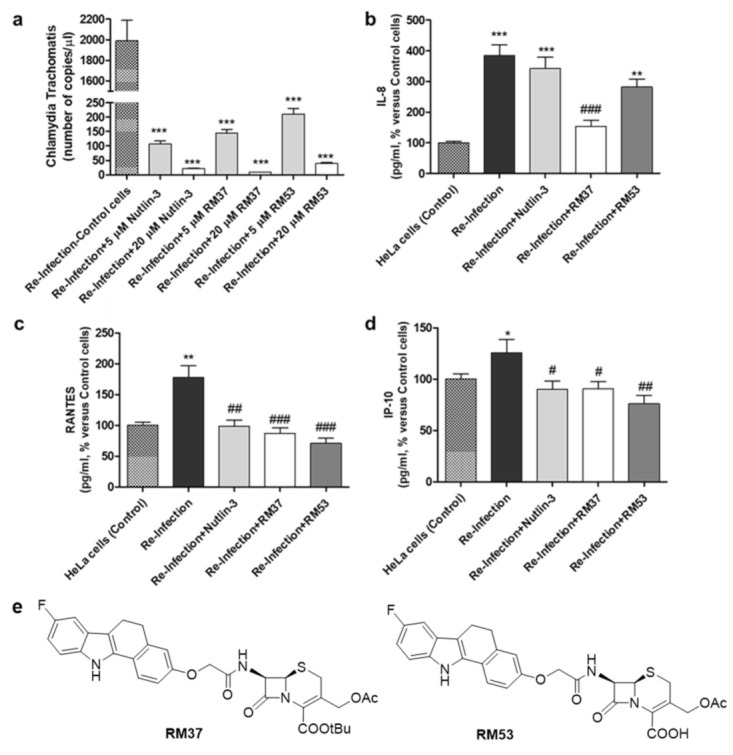
Quantification of *Chlamydia trachomatis* on HeLa-infected cells. HeLa cells were infected with *Chlamydia trachomatis* and then treated with Nutlin-3a, **RM37**, and **RM53**. (**a**) Following reinfection, real-time PCR analysis was performed to quantify *Chlamydia trachomatis* copies on untreated (Reinfection) and treated infected cells. Data are expressed as the number of copies/µL in each condition. (**b**–**d**) Following reinfection, supernatants of HeLa cells were used to quantify the indicated cytokines and chemokines by Luminex assay. Data are expressed as pg/mL in each condition, versus those of control cells, and are mean ± SEM of three different experiments. * *p* < 0.05, ** *p* < 0.01, *** *p* < 0.001 versus control. ^#^
*p* < 0.05, ^##^
*p* < 0.01, ^###^
*p* < 0.001 versus reinfection. (**e**) Chemical structures of hit compound **RM37** and its derivative **RM53**.

**Figure 10 pharmaceuticals-15-01318-f010:**
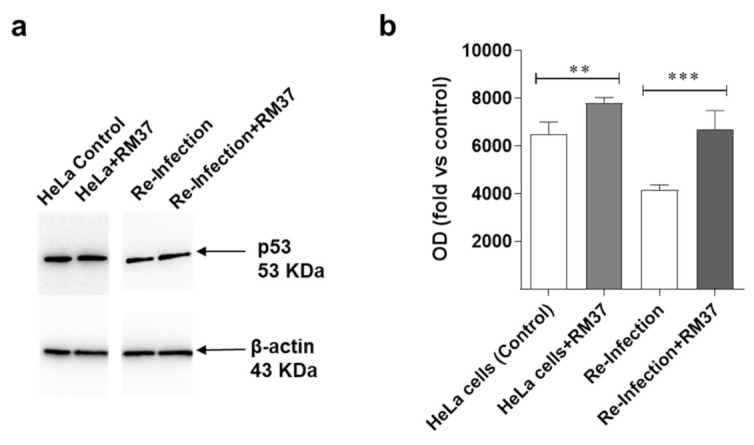
Effects on p53 pathway’s reactivation upon C. trachomatis infection. (**a**,**b**) HeLa cells were inoculated with C. trachomatis serovar D (ATCC VR-885) at multiplicity of infection (MOI) 0.5 and treated with **RM37**. Cell lysates were subjected to Western blot analysis using specific antibody for p53. (**a**) A representative image is shown. Full screen of Western blot assay was provided in a separate file. (**b**) Data are shown as arbitrary optical density (OD) (mean values ± SEM of three different experiments). ** *p* < 0.01, *** *p* < 0.001 versus control.

**Table 1 pharmaceuticals-15-01318-t001:** Structures, in-house database codes, and MDM2 inhibitory activity of compounds selected from VS. ^a^ Evaluated by in vitro p53-p/HDM2 binding assay, as described in Costa et al., 2013 [25].

Code	Compound Structure	% Inhibition ^a^ at 20 µM
**RM37**	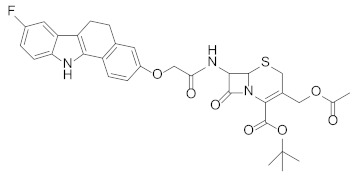	76.2
**RM58**	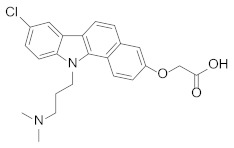	70.5
**RM43**	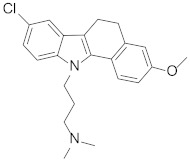	<10
**RM45**	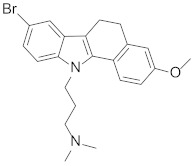	<10
**RM82**	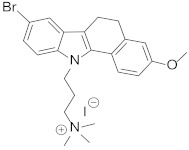	<10
**SN59**	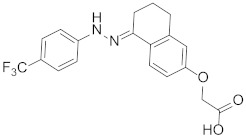	<20
**Nut-3a**		100

## Data Availability

Data are contained within the article and Appendix A.

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
