# Peer review of "Identification of a Novel p53 Modulator Endowed with Antitumoural and Antibacterial Activity through a Scaffold Repurposing Approach"

_pharmaceuticals, 2022, doi:10.3390/ph15111318_

Round 1

Author Response

Comments:

The article is well written and in a sequence that attracts the reader. The idea of the research is innovative, and I think it will be a highly citation paper.

Below are comments focused on improving the results and others that can be useful in showing the results to the reader better.

  1. For Figure 3, please provide the full proton spectrums (could be added in the same picture or in supplementary)

Reply:

The full proton spectra for Figure 3 have been added in Supplementary Information as Figure S1.

  1. Line 153: it stated that “The data are the mean values ± SEM of two different experiments, each performed in duplicate”. The experiments should at least perform in three experiments each performed in triplicate.

Reply:

As required by the Reviewer, additional replicates of the experiment were performed. The data are depicted in the new Figure 4 and reported in the text.

  1. Line 177 and Line 194: figure 5 (a) and Figure 6 (b) the full screen of the westernplot assay must be provided.

Reply:

As required by the Reviewer, the full screen of the western blot assay were provided in a separate file as Supporting Information. This issue was specified in the legend of Figure 5a and Figure 6b.

  1. Line 321: it stated that “The data are the mean values ± SEM of two different experiments, each performed in duplicate”. The experiments should at least perform in three experiments each performed in triplicate.

Reply:

As required by the Reviewer, additional replicates of the experiment were performed. The data are depicted in the new Figure 10 and reported in the text.

  1. Line 245 to 251: for the cell cycle experiment, you should indicate the actual value (% of cell phase) in the texts.

Reply:

As required by the Reviewer, the percentages of cell phase were reported in the text.

  1. Line 263: it stated that “The data are the mean values ± SEM of two different experiments, each performed in duplicate”. The experiments should at least perform in three experiments each performed in triplicate.

Reply:

As required by the Reviewer, additional replicates of the experiment were performed. The data are depicted in the new Figure 8 and reported in the text.

Reviewer 2 Report

for novel MDM2 inhibitors able to restore p53 activity. Authors performed a virtual screening using a chemical library composed earlier by small molecules synthesized for different targets. The most promising compounds were tested in vitro for inhibition of MDM2 activity. This led to selection a hit compound RM37 which was proved to ability to restore p53 function.

The Authors are requested to comment on some issues and make corrections:

1. Two compounds, RM37 and RM58, showed inhibition of MDM2 more than 70% (lines 105-108). Why the NMR studies were performed only for one compound?

2. Figure 4, 5, 6, 7, 8, 9 - Authors should shorten the explanation for these figures. Some sentences should be included in the text of manuscript.

3. The chemical structure of RM53 should be introduce to the manuscript.

4. The 1H NMR analysis (Supplementary) should be improved, each of hydrogen atoms being at a specific position should be assigned to observed signal(s).

Author Response

Comments:

  1. Two compounds, RM37 and RM58, showed inhibition of MDM2 more than 70% (lines 105-108). Why the NMR studies were performed only for one compound?

Reply:

The NMR binding studies were initially performed only for the most active compound, RM37, showing a 76.2% inhibition of MDM2 at 20 µM concentration. The other potential hit compound, RM58, showed a slightly lower potency in the in vitro binding assay (70.5% inhibition of MDM2 at 20 µM) but it was much less active than RM37 on U343MG cell proliferation (Figure 4a,b). For these reasons, all further studies were focused on RM37 as hit compound.

  1. Figure 4, 5, 6, 7, 8, 9 - Authors should shorten the explanation for these figures. Some sentences should be included in the text of manuscript.

Reply:

As required by the Reviewer, the explanation of the figures was shortened including some sentences in the text.

  1. The chemical structure of RM53 should be introduce to the manuscript.

Reply:

For clarity reasons, the chemical structure of the carboxylic acid RM53 originally present only in the Supporting Information has been now added in Figure 9e of the revised version of the manuscript.

  1. The 1H NMR analysis (Supplementary) should be improved, each of hydrogen atoms being at a specific position should be assigned to observed signal(s).

Reply:

The 1H NMR analysis of RM37 and RM53 in the Supplementary material have been improved, each signal has been assigned and a HSQC spectrum for RM37 has been added.